# Reengineering Human Resources and Entrepreneurial Learning towards Organizational Revitalization in Malaysian Travel and Tourism Companies during the COVID-19 Pandemic

Cheng-Wen Lee [1,*] , Choong Leng Peng [2] and Hsiao Chuan Chen [2]

1    Department of International Business, College of Business, Chung Yuan Christian University,
     Taoyuan City 320, Taiwan
2    Ph.D. Program in Business, College of Business, Chung Yuan Christian University, Taoyuan City 320, Taiwan
*    Correspondence: chengwen@cycu.edu.tw

**Abstract:** The interest in the reengineering process of human resources and entrepreneurial learning contributes to enhancing the success of organizational revitalization, especially concerning corporate sustainability. This study builds a conceptual research framework and adopts the descriptive analytical approach based on the data of 239 samples from the employees working in travel and tourism companies in Malaysia. The findings indicate the existence of a statistically significant effect of human resources reengineering on various dimensions of organizational revitalization. Amongst them, pioneering/entrepreneurial learning as a moderating variable also has a significant and positive influence on organizational revitalization. The implications of this study mostly emphasize the necessity for travel and tourism companies to maintain their sustainability through reengineering human resources. In accomplishing strategic tasks for human resource development and achieving the strategic goals, these essential issues such as encouraging workers to innovate and continuously raising their awareness of entrepreneurship need to be acknowledged.

**Keywords:** COVID-19; human resource management; entrepreneurship culture; reengineering human resources; entrepreneurial learning; organizational revitalization

## 1. Introduction

Since people witness successive crises that relentlessly rock global economies, a set of radical and accelerating changes occur, especially due to tremendous development in production globalization and information technology. The coronavirus (COVID-19) pandemic is casting its shadow over all business sectors and already causing many industries large-scale losses and imbalances in their future plans. Organizations and companies must search for efficient ways to save and revive themselves and explore new markets and resources to increase market share, improve operations and create high flexibility and speed to react to business environmental impact. In order to maintain the continuity of services, companies need to effectively cope with the remarkable transformation in the patterns of customer behaviors and purchasing/consumption cultures.

One of the sectors most affected by the COVID-19 pandemic is the tourism sector, as many tourists cancel trips due to the closure of some destinations by the local government, which in turn brings disadvantages to land, sea, and air transportation because of a significant decrease in traveling passengers. However, other sectors also have the potential to benefit from the COVID-19 pandemic, including the agriculture sector, personal healthcare and medical services, and e-commerce [1]. The fear of infection with the COVID-19 pandemic, the uncertainty of jobs and careers, and the negative information circulating through social media further deteriorate the tourism sector economy. Additionally, with the implementation of "work from home" (WFH) for employees in numerous private and government companies, tourism companies face several rapid bottlenecks to overcome. The

COVID-19 pandemic has repercussions on tourism companies more than its predecessors such as viruses and epidemics that previously spread or other concerns [2].

The seriousness of the fact that the tourism sector is easily affected by global crises calls for more attention because this sector represents economic and commercial importance for many countries and constitutes large percentages of their Gross Domestic Product (GDP). In 2000, the International Labor Organization revealed the need for urgent attention to the devastating situation caused by the COVID-19 pandemic in the tourism sector. In addition to the COVID-19 pandemic, the issue of travel insurance has come to the fore, which may be a way to revive this sector by providing travel packages with travel insurance services that may reduce the concerns that threaten this sector. Given the high level of the importance of tourism as a contributor to GDP and a major source of income and jobs, this pandemic affects economies and labor markets due to international travel restrictions and procedures. The closures imposed by governments in many countries and the recommendations related to staying at home also deal a blow to domestic tourism, which will lead to a contraction and stunted growth in the economies of many countries, especially Asian ones.

The next stage, after working to mitigate the spread of the COVID-19 pandemic, must include supporting and enabling future activities by defining new human resource strategies that determine the value of human capital in adapting to the future and new expectations. The understanding of influential human forces in the current market supports the formation of human resources strategies. The transition to the reengineering and reconfiguration of human resources provides a clear roadmap for utilizing human capital in reviving both organization's goals and challenges [3]. However, organizational revitalization is a complex process that requires many changes, capabilities, and solutions, needs to be implemented and accepted by the institution and stakeholders, and eventually brings organizational value, vitality, and enthusiasm [3]. An organization puts a lot of effort into adapting to internal and external pressures from reengineering human resources. It is a very powerful global formula for changing the culture, whether in an entire community or in an organization, for threatened groups. The process of revitalization must include answering several questions, such as the timing and motivation of process regeneration, interesting aspects of implementation, the important part of innovation, the prepared period of organization, and so on [4].

In light of the abovementioned, the study aims to test the relationship of the three variables, including the impact of reengineering human resources on organizational revitalization and the existence of pioneering/entrepreneurial learning as a moderating variable. We use the travel and tourism companies in Malaysia as our study sample because these companies are the most in need, at present, of organizational revitalization due to being affected by the repercussions of the COVID-19 crisis.

## 2. Literature Review

### 2.1. Organizational Revitalization

The experience of organizations suggests that the new relationship between personnel and line managers has to work during the initial phases of reengineering human resources (RHRs), and then the line managers should be considered efficient enough to take on greater responsibilities related to personnel management activities. Eventually, organizations should evaluate and review performances and activities [5]. One of the most important pillars of the process of organizational revitalization is the human resource component, as organizations that wish to continue their work and develop depend on employees with specialized knowledge and technological skills who are creative, enthusiastic, and unconventional [4]. Organizational revitalization clarifies a collaborative relationship between managers and scientists to improve performance; that is, the importance and the primary role of this process rests with human resources [6]. All large companies have gained access to rare and unreplaceable resources of strategic strength, technology, and capital, which have become human and organizational capacity that can be mobilized to the company's unique competitive advantage better than competitors' achievements [7].

In order to address these problems and implement the organizational revitalization processes in the affected companies, we need many fast and effective tools/methods that contribute to saving companies from the repercussions of the damaging effects that may increase over time and achieve the highest compatibility between intellectual and human capital. The most important thing is the restoration of organizational RHRs. Reengineering is generally known as a rapid and radical redesign of the processes, services, policies, and organizational structure of an organization. Reengineering is also the art of changing the way the organization thinks and thus does things in a radical way [8]. RHRs are transforming human resource skills, capabilities, and knowledge into intellectual capital and achieving the goals of the organization and relevant parties [9]. The comprehensive review of critical human resources contributes to raising the efficiency of the organization's work and enhancing its ability for higher quality services [10]. Through these definitions, the contribution of strong reengineering human resources can be clearly evident in light of the availability of the organizational ingredients in the process of organizational revitalization.

*2.2. Reengineering Human Resources*

Reengineering has been used to refer to a wide range of organizational changes, including downsizing, restructuring, and process improvement [11]. Reengineering human resources (RHRs) means radical rethinking and the role of new technology in the optimization of human resource management (HRM) [5]. Reengineering is not down-sizing—it eliminates work, not jobs—it is not HR restructuring—moving boxes on an organizational chart—it is not automation, and it is not reengineering a department, but a process in an organization [12]. As the environment has become more volatile and talent (in many forms) is more central, strategically, HRM is pushed for more strategic results while still maintaining all the operational services needed by employees. Christenson [13] and Boudreau and Ramstad [14] encourage and support the transition of HRM into more strategic arenas of the organization. An empirical result found that the contribution of RHRs provided and strengthened high-performance systems or practices on the basis of surveying the opinions of managers in some economic institutions in Iran [15]. A questionnaire survey as a study tool was distributed to workers at higher administrative levels in seven service and industrial economic institutions in Iran. The most important result was the existence of a statistically significant effect of RHRs on strengthening high-performance work systems in economic institutions, where the explanatory capacity was acceptable, as RHRs explained 21.3% of high-performance work systems [15].

Students' attitudes towards a pioneering experiment in teaching and learning methods of the College of Administrative Sciences and Information Systems at Palestine Polytechnic University, which the college has followed since 2016, measured the trend towards interactive activities with society and identified whether there were statistically significant differences in attitudes [16]. Due to the variables of the academic year, gender, and specialization, and to achieve the objectives of the study, the researcher adopted the descriptive method and used a questionnaire as a tool to collect data from 258 students of Palestine Polytechnic University as the study sample, who were chosen by a random stratified method [16]. Both the students' attitudes towards pioneering teaching and learning methods and their attitudes towards interactive activities with institutions and society were highly positive. Additionally, the results showed that there were no statistically significant differences in the students' attitudes due to the variables of the academic year, gender, and specialization towards entrepreneurial learning [16].

The effect of entrepreneurial learning on the relationship between attitudes towards entrepreneurship, self-standards, and behavior control is significant [17]. To achieve the goal of the research, the researcher adopted the relational descriptive approach, and the study sample consisted of 500 students of Ain Shams University. The results showed the presence of a statistically significant effect of entrepreneurial learning on trends towards entrepreneurship, self-standards, behavioral control, a tendency to take risks, and a love of achievement with entrepreneurial intentions. The concept of entrepreneurial learning and

its positive benefits, in addition to providing an understanding of entrepreneurial learning for women entrepreneurs in the creative industry, was researched [18]. To achieve the objectives of the study, the researchers adopted the inductive and qualitative approach, which relies on reviewing the previous literature related to entrepreneurial learning, in addition to conducting interviews with 20 workers in Essex, UK. Entrepreneurial learning is a dynamic process that occurs through the interaction between the personal and social experience of the entrepreneur during the entrepreneurial journey, which is independent of the unique context of the creative industry, and the learning procedures for creative entrepreneurs were characterized by personal experiences coupled with a high level of cooperation and participation. In short, entrepreneurial learning affected the entrepreneurial efficiency of women entrepreneurs [18].

RHR tools in the context of crisis control development strategy in corporate structures, as well as the impact of RHR processes on reducing the crises faced by industrial companies in the Netherlands, were researched [19]. The researchers adopted an inductive, qualitative approach based on reviewing the literature related to the subject of the study as well as obtaining information, data, and performance and financial indicators from industrial companies in the Netherlands and analyzing those data and indicators. The study acquired several results. During the application of reengineering human resources, many complex factors in the workers' companies are taken into account, related to the business process, budget, organizational structure, and workplace system. Reengineering human resources is a central element in the strategy of developing institutional structures to face crises.

The reality of entrepreneurship culture and the mechanisms of its activation at Suez Canal University in Egypt from the viewpoint of 520 students of the Commerce and Engineering Department and the importance of entrepreneurship education to develop skills and competencies was highlighted [20]. Entrepreneurship culture can enhance competitiveness in the labor market, in addition to clarifying the role of entrepreneurship education in achieving sustainable development. In summary, there is a correlation between entrepreneurial education and sustainable development, and entrepreneurship education for innovation, risk, and creative ideas has a role in providing job opportunities.

Innovation or reengineering can stimulate organizational growth, develop innovations for adapting the organization to a new culture or era, and provide institutional support for high-tension faith in the organization [21]. Creativity plays a critical role in the ability of organizations to thrive long-term. Rapid advances in technology and stakeholder proclivities can cause uncreative organizations to become stagnant and eventually die. Leaders can orchestrate revivals, however, with the reintroduction of creativity into organizations. To accomplish this, CEOs or managers should ensure that the organization's structures and systems are conducive to creative processes, model creative behavior, and encourage ongoing creative activity. In doing so, leaders produce thriving, competitive organizations which are adept at creative thinking, nimble in times of crisis, and which have continuous, robust pipelines of creative ideas from which to draw [22].

### 2.3. Entrepreneurial Learning

To activate the relationship between RHRs and organizational revitalization, several studies have suggested the use of entrepreneurial learning, which is known as learning to recognize and act upon opportunities and to interact socially to initiate, organize, and manage projects [23]. Entrepreneurship courses at the higher education level are organized courses that contribute to the development of knowledge, skills, and entrepreneurial positions to enhance competencies, which increase the performance of the company [24]. Furthermore, the company's survival and long-term performance are related to the competencies of entrepreneurs [25], and the company's performance, growth, and profitability raise the entrepreneurial competencies [26]. To describe entrepreneurial learning, it is the process of identifying opportunities and focusing on the entrepreneur's cognitive mechanisms for identifying business opportunities and making decisions about them [27]. A number of authors have recognized that entrepreneurial learning is not only related

to obtaining information but also the acquisition of a variety of human and non-human resources [28]. Therefore, this study concludes that entrepreneurial learning is a continuous process that facilitates the development of the knowledge necessary to be effective in initiating and managing new projects and processes of organizational revitalization and the advancement of the organization. Entrepreneurial learning is also called institutional education, which seeks to enhance self-esteem and confidence by relying on one's talents and creativity while building skills and related value in expanding horizons for education and exploiting opportunities. The role of professional expertise, the transformation process, and entrepreneurial knowledge in promoting entrepreneurial learning have been described [29].

Hiroshi Mikitani, founder and CEO of Rakuten, Japan's biggest e-commerce retailer, offered some guidance on how to fix a broken company [30]. Leaders have to make the problems clear [31]. What is "troubling" the organization? Where is it happening? Does management have all the facts related to the trouble at hand? Has a senior management team validated the facts? If facts are not available, then a small team needs to be assigned to get the facts right. A failing organization faces multiple problems that need to be addressed, such as quality, features, processes, communication, roles and responsibility, accountability, people, transparency, delay in making decisions, challenging the status quo, super dominant characters, etc. Once organizations have identified and sorted the problems in different buckets identify the right people (a task force) within the corresponding departments/organization to fix the problems. Please note—don't go for an external consultant to come and fix the problem within these buckets reason being they know less of everything about you and your problems. Once the right people are identified—figure out if they are motivated to fix the problems in their corresponding areas. Get them all to a room and give a super charged speech to get them excited and motivated. Do not dictate people on "How to fix it?" The group knows and will find a way to fix the problem. Give them the freedom to change rules/processes in this step. Track the progress—Set up milestones and set some realistic expectations. In order to evaluate progress each working team has to identify few simple key metrics based on data points to compare in the future. Please note the selected key metrics has to be something easily quantifiable and simple for the team to be measured against [30]. For example, selected metrics can be something such as the current product defect ratio coming out of the manufacturing pipeline, the current customer satisfaction index score, or the overall time it takes for a customer to check out items bought on the company website. As much as possible, refrain from coming up with complicated evaluation criteria.

Going to the final step, each individual task force working on fixing problems within a particular bucket needs to be constructively challenged and provided with the right support from top management. When challenged, the team obtains a better understanding of the problems at hand, which helps them to think clearly, clarify strategies, and ideas are refined. Some questions to start with are: Why does the team think this strategy might work? What is the difference between this strategy and the previous one? What assumptions are made in arriving at the strategy? Is there field data to support the assumptions? Set all stakeholders' expectations for positive results anywhere between 6 and 18 months. Evaluate the results—at periodic intervals, evaluate each team's key metrics and compare them against the last recorded values. So, after six months, if the product defect ratio has reduced from 4% to 2%, the customer satisfaction index score has increased from 80 to 85, and the checkout time has reduced from 100 s to 90 s. Periodic results evaluation will let you know what areas are improving and at what rate and what areas are not. Based on the achieved results, individual teams can be tasked to do better on the selected metrics that lack the desired progress. The corresponding team can revise its strategy to address the particular metric.

## 3. Research Conceptual Framework

Companies all over the world, especially after the COVID-19 pandemic, for which everyone is waiting for the end of its consequences, seek to remedy the global crisis situation and avoid the risks surrounding them that threaten their survival and existence by searching for new strategies and directions that save them from the clutches of tampering in the economic system. To continue growth, achieve profits, or maintain stability—at a minimum to stop losses and maintain survival—are very crucial in the global economy. The need for revitalization and revision has become a necessity with the emergence of new human sciences, increasing aspirations, specialization, and professionalism, and the increasing need for interdependence in light of more turbulent and less predictable external environments [32]. It is also necessary to achieve this to understand the factors that lead to a decrease in the level of vitality in the organization and the impact on the productivity of the employee [4].

Since the World Health Organization declared COVID-19 as a pandemic in March 2020, the tourism and travel sector (e.g., hotels, airlines, cruises, and car rentals) has witnessed a significant decline, with long-term negative expectations [33]. The industries in this sector have made structural changes that cannot be rejected or resisted. The transformation of the organization with the purpose of creating or adding value is the essence of the process of organizational revitalization, and there is little coverage in the literature provided for the costs of developing and regenerating the organization and determining the optimal levels of such. There is justification for the need for additional research on the topic of how to control entrepreneurial learning processes and promote the process of the systemic revitalization of certain divisions [34].

It appears from the foregoing that the problem of the study is summarized in the existence of a theoretical knowledge gap and a clear field gap that focuses on the limited interest and focus on the processes of organized revival and the lack of sufficient understanding to deal with it, despite the presence of a desire, willingness, and limited practices for it in the researched companies, in addition to losing the opportunity of its relationship to RHRs and entrepreneurial learning.

### 3.1. Independent Variable

The independent variable is the radical and complete review of the human resources in the organization to enhance its activities and make them more efficient and more able to provide higher quality and stimulate innovation [10]. Procedurally, it is defined as the process of re-employing and investing human resources in the organizations in Malaysian tourism and travel companies, and it will be measured by the degree of the response of the sample members to their paragraphs on the scale. We discuss the RHRs from four dimensions.

(1) Reengineering work culture: Reshaping the personality of the organization, which is embodied in the common understanding of the organization's mission, values, decision-making, activities at all levels, the relationship of subordinates with their managers and vice versa, and enhancing the sense of membership and belonging and the exchange of ideas within the organization and its sustainability [35].

(2) Reengineering training and development: The process of improving and adding value to incomplete human resources in order to perform their new roles, so that learning is provided to employees while they are on the job to enhance their levels of knowledge and skills, and this task and its responsibilities are shared by training and development, direct managers, and human resources [5].

(3) Reengineering business processes: A radical reconstruction and restructuring of processes in order to achieve fundamental improvements for the company that include critical areas for economic growth, the main motivation of which is to ensure the development of the company's structure to face the crisis, and includes composition and structure of jobs–duration–cost–size of products or services–quality. In connection with that, reengi-

neering means changing the basic principles of a company or organization and focusing on operations and not on jobs [19].

(4) Reengineering workplace systems: Re-developing and organizing the workplace as an organizational approach through a system that uses visual cues and visual management to organize the workplace to help reduce some forms of waste and show problems sooner, so that each element is in the right place and position in WMS. It has a 5S to launch its "Operational Efficiency Journey" [36]. Based on the theoretical arguments, we posit Hypothesis 1, covering H1a, H1b, H1c, and H1d.

**H1:** Reengineering human resources (RHRs) significantly and positively affect the organizational revitalization of travel and tourism companies.

**H1a:** Reengineering work culture significantly and positively influences organizational revitalization.

**H1b:** Reengineering training and development significantly and positively influence organizational revitalization.

**H1c:** Reengineering business processes significantly and positively influence organizational revitalization.

**H1d:** Reengineering workplace systems significantly and positively influence organizational revitalization.

### 3.2. Moderating Variable

The moderating variable is a continuous process of development, training, and education for managerial and work skills (business skills, management, communication, leadership, problem-solving and decision-making, teamwork, initiative, ambition, drive, and achievement) and the promotion of a culture of creativity and innovation to meet the organization's needs for the knowledge and behavioral skills necessary for the labor market, with the aim of enhancing the growth and productivity of the organization [37].

(1) Professional experience: The set of special knowledge and skills acquired during practice or through experience, in addition to the skills, competencies, and innate observation strength that achieve a high level of performance and reduce expenses [38].

(2) Transformation process: In entrepreneurial learning, it is an experiential process through which personal experience in entrepreneurship is transformed into knowledge, which in turn can be used to guide the selection of new experiences [29]. It is an activity or group of activities to create new outputs and add value [39].

(3) Entrepreneurial knowledge: Information and experiences that entrepreneurs gain about contacts and relationships with people and stakeholders involved in the company's business, potential new viable markets, product availability, resources, and competitive response, which enhances the ability to respond to change and seize and discover entrepreneurship opportunities, and dealing with obstacles, problems and technological developments in different ways to achieve greater success [29]. Therefore, we propose Hypothesis 2.

**H2:** Entrepreneurial learning is significantly related to moderating the impact of RHRs on the organizational revitalization of travel and tourism companies.

### 3.3. Dependent Variable

The dependent variable is the number of proposed changes that have been successfully implemented to give additional value and energy to the company and achieve a high level of innovation that raises the level of performance in times of crisis and serious investment in details to make a difference, in light of the increasingly fierce competition due to globalization and technological development. This process requires a high degree of awareness and an early assessment of the state of the company and the will to change by management [4].

(1) Working atmosphere: Every part of the employee's participation in the work itself, such as relationships with colleagues and bosses, organizational culture, and space for self-development. It is divided into two types. The physical work environment is represented in all the physical conditions around the workplace and affects employees in a direct or indirect way. Physical factors include "workplace air temperature, work area, noise, density, and area". The other type (non-material work environment) includes factors that are not visible and can be felt, such as the relationship between colleagues, bosses, and subordinates and values and trends prevailing in administrative companies, in which there is a healthy work climate that improves productivity in the company [40].

(2) Knowledge making: It is defined as maximizing intellectual capital and information and measuring it in modern technological ways, with the aim of reusing it to create new value through improving efficiency and effectiveness among individuals, and cooperation to enhance innovation and decision-making so that its outputs are used to gain market share and achieve a competitive advantage by arranging the accumulated ideas and experiences using advanced technology [41].

(3) Futurology: An organized scientific effort of long-term planning, forecasting, and the anticipation of future developments by investing in previous experiences, knowledge, and information and monitoring future changes and developments, especially in the field of technology, arising as a result of rapid developments in the world and aiming to reach an advanced stage of knowledge of future trends. The goals are to exploit opportunities, avoid threats, and find alternatives and solutions [42]. In accordance with the literature and Hypotheses proposed, this study establishes a conceptual research framework (Figure 1).

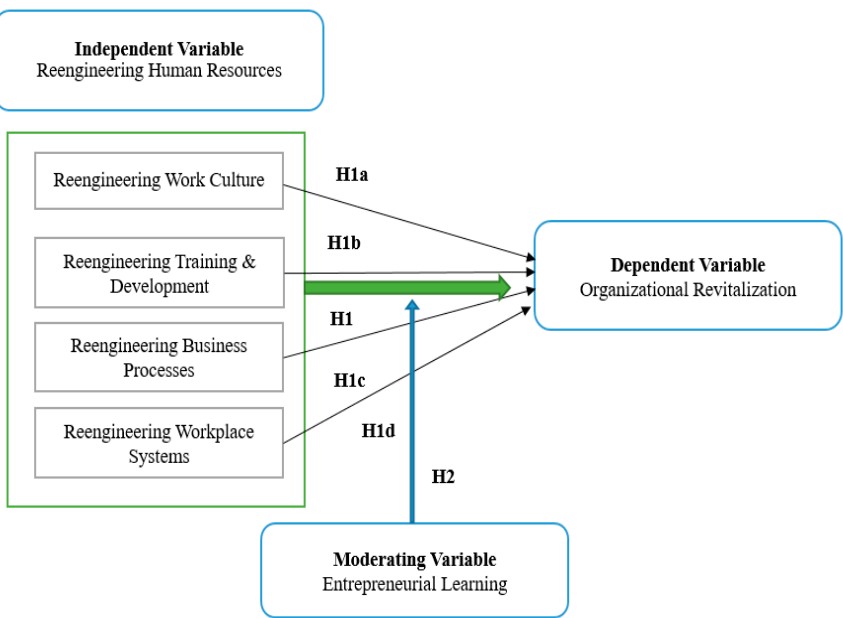

**Figure 1.** Conceptual research framework.

## 4. Methodology

### 4.1. Data Collection

The present study is a causal study with a quantitatively thorough questionnaire (see Appendix A). It adopts the descriptive analytical approach method that is associated with describing the results with analysis. The field of study consists of travel and tourism companies in Malaysia that are registered with the Ministry of Tourism, Arts, and Culture (2021), whose financial guarantee amounts to USD 25,000. There are 63 companies, containing three administrative levels—higher, central, and supervisory. As for the study population, this may consist of all workers in the tourism and travel companies within the aforementioned levels (upper, middle, and supervisory management), who number

768 employees of the Malaysian Association of Tour and Travel Agents. The result is 239 valid responses (31.1%).

The questionnaire was circulated in a Google form with the help of different means of social media. Most of the data were collected through a Google Excel sheet which was further analyzed and interpreted. Regarding the supervisory level, e.g., CEOs or managers, we contacted them by telephone and requested them to take part in this survey. On agreeing to do so, each CEO or manager was mailed an anonymous questionnaire together with a cover letter and a stamped, addressed envelope (or email address) for its return. The design of the questionnaire facilitates the respondent sample by providing a data source for other research topics unrelated to this one, thereby allowing significant research efficiency.

To evaluate the relative importance of different question items, each respondent specified the relative importance of characteristics of RHRs, entrepreneurial learning, and organizational revitalization on a 5-point Likert scale, indicating whether relative importance is very low (1) or very high (5). In this study, the method of reliability measurement is Cronbach's coefficient $\alpha$. Cronbach's coefficient $\alpha$ was calculated for each of these factors to assess the internal consistency of the model constructs. According to Price and Mueller [43], a standard coefficient $\alpha$ of 0.60 or higher is generally considered acceptable when using a measure. If statistical significance is not achieved, the research may need to eliminate the indicator or attempt to transform it for a better fit with the construct. Here, the values of Cronbach's coefficient $\alpha$ of all variables exceeded 0.80. This indicates that the research has good consistency and stability.

### 4.2. Structured Equation Model

To test these hypotheses (H1a, H1b, H1c, H1d), we adopted the Structured Equation Model (SEM) method through the AMOS program. The manifest variables collected from the respondents are indicators of the measurement model, as we used them to measure or indicate the latent constructs. The most obvious difference between the measurement model and factor analysis is that the former has a much smaller number of loadings and resembles the exploratory mode of actor analysis. Researchers can specify a measurement model for both exogenous constructs and endogenous constructs.

In order to specify the measurement model, this study makes the transition from factor analysis, in which the researcher has no control over which variables describe each factor, to a confirmatory mode, in which the researcher specifies which variables define each construct (factor). The latent variables include reengineering work culture (RWC), reengineering training and development (RTD), reengineering business processes (RBP), and reengineering workplace systems (RWS). The model fit assessment approach is involved, using several diagnostics to judge the simultaneous fit of the measurement and structural models to data collected for this study.

### 4.3. Multiple Hierarchical Regression Analysis

Regarding H2, we used the multiple hierarchical regression analysis to demonstrate the impact of entrepreneurial learning on modifying the impact of human resources reengineering dimensions (reengineering work culture, reengineering training and development, reengineering business process, and reengineering workplace systems) on the organizational revitalization of tourism companies in Malaysia. Multiple regression analysis is a type of test that analyzes the amount of variance explained in a dependent variable by more than one predictor variable. A hierarchical multiple regression analysis adds another piece, in that independent variables are entered in blocks. The test allows authors to look at the $R^2$ change and F-statistic change between the two models, in addition to reporting the level of significance for each one.

## 5. Results

### 5.1. Characteristics of Respondents

The study sample is a proportional stratified random sample with a size of 239 individuals in Malaysian tourism and travel companies who constitute all three administrative levels (higher, middle, and supervisory), which is a size that represents the study population properly. The characteristics of the respondents are illustrated in Table 1. Table 1 shows that the percentage of males is higher than the percentage of females in the study sample, where the percentage of males is 53.6% and the proportion of females is 46.4%. Although the percentage of males is higher than the percentage of females, the difference between the ratios of males and females was not found, owing to the nature of the work of tourism and travel companies that depend on both males and females in their work. The age-related data of the table also indicate that 34.7% of the study sample were within the age group of less than 30 years, which is the highest percentage within this sample, followed by those who were included in the age group of 31 to less than 40 years, at a percentage of 21.6%, followed by those in the age group of 41 to less than 50 years at a proportion of 19.2%, followed by the age group of 51 to less than 60 years with 20.1%, followed by those in the last place in the age group of 60 years and over. This indicates that tourism and travel companies are attracting the young, who are characterized by energy, motivation, and the ability to adapt to the surrounding environment and the diverse cultures necessary to perform the work of these types of companies operating in the tourism sector. With regard to the educational level, it is noticed that holders of bachelor's degrees are the most represented group within this sample, constituting 40.6%, followed by those within the level of general secondary or community diploma at a rate of 24.7%, followed by those at the master's level at a rate of 20.9%, followed by those within the educational level of higher diploma at a rate of 8.8%, and in the last place are those within the level of a doctoral degree at a rate of 5.0%. This result indicates that most workers in travel and tourism companies are holders of bachelor's and master's degrees, as they formed a combined rate of 61.5%, and this indicates that the majority of businesses in travel and tourism companies require higher degrees and attract those within the educational levels of bachelor's and above, in addition to the educated nature of Jordanian society.

**Table 1.** Characteristics of respondents (*n* = 239).

| Variables | Categories | Frequency | Ratio (%) |
|---|---|---|---|
| Gender | Male | 128 | 53.6 |
| | Female | 111 | 46.4 |
| Age | 30 years or less | 83 | 34.7 |
| | 31—less than 40 years | 51 | 21.6 |
| | 41—less than 50 years | 46 | 19.2 |
| | 51—less than 60 years | 48 | 20.1 |
| | 60 years and over | 11 | 4.4 |
| Educational level | High school or community diploma | 59 | 24.7 |
| | Bachelor's degree | 97 | 40.6 |
| | Higher diploma | 21 | 8.8 |
| | Master's degree | 50 | 20.9 |
| | Doctoral degree | 12 | 5.0 |
| Work position | Director general | 25 | 10.5 |
| | Deputy general manager | 18 | 7.5 |
| | Head of the department | 101 | 42.3 |
| | Employee | 95 | 39.7 |
| Years of experience | 5 years or less | 76 | 31.8 |
| | 6~10 years | 82 | 34.3 |
| | 11~15 years | 46 | 19.2 |
| | 16 years and over | 35 | 14.7 |

As Table 1 shows, the majority of the sample members are heads of departments (42.3%) and employees (39.7%), while general managers formed 10.5% and deputy directors accounted for 7.5%. This result is explained by the nature of the organizational and functional structures of the travel and tourism companies, as these structures may include one director followed by many heads of departments and employees. Finally, the table data indicate that 31.8% of the study sample have 5 years of experience or less, followed by those with 6 to 10 years of experience (34.3%), followed by those with 11 to 15 years of experience (19.2%), and in the last rank are those who have 16 years of experience and over (14.7%). This result is explained by the concentration of most of the sample members with low years of experience due to the nature of the age groups of the sample, which concentrates on the young group.

*5.2. Descriptive Statistics*

The estimates of the sample members are identified by computing the arithmetic means and the standard deviations of their answers, in addition to the value of the (t) test, and the results are shown in the following tables. It can be noticed from Table 2 that the average estimates of the sample members about the relative importance of the dimensions of re-engineering human resources ranged between 3.83 and 4.05 and came after re-engineering work culture in the first place with the highest arithmetic mean of 4.05 and with high relative importance. This is followed by the dimension of re-engineering business processes, with an arithmetic mean of 3.90, and of high relative importance; followed by the dimension of re-engineering workplace systems, with an arithmetic mean of 3.86 and high relative importance; and in the last place is the dimension of re-engineering workplace systems with an arithmetic average of 3.83 and high relative importance.

**Table 2.** Reengineering human resources dimensions.

| Dimensions | Arithmetic Mean | Standard Deviation | (t) Value | Materiality |
|---|---|---|---|---|
| Reengineering work culture (RWC) | 4.05 | 0.59 | 25.44 | High |
| Reengineering training and development (RTD) | 3.83 | 0.67 | 17.91 | High |
| Reengineering business processes (RBP) | 3.90 | 0.68 | 19.16 | High |
| Reengineering workplace systems (RWS) | 3.86 | 0.73 | 17.09 | High |
| Overall average | 3.91 | 0.54 | 24.29 | High |

The data of the previous table also indicate that the arithmetic mean of the sample estimate on the dimensions of reengineering human resources as a whole reached 3.91 with high relative importance, meaning that there is a high level of reengineering human resources in the travel and tourism companies operating in Malaysia. The value of the (t) test was 29.24, which is a statistically significant value as it is greater than the tabular value of 1.96. The results of the preceding indicate that tourism companies, the study sample, consider the physical work environment influencing the productivity of the work, so they adopt the flexible interior design of the workplace, provide offices that meet the requirements of performing the tasks, design the building spaces in a manner consistent with the performance of tasks, and prepare the necessary logistics and the workplace with ease, which made the responses of respondents tend to have a high degree of approval.

It can be noticed from Table 3 that the average estimates of the sample members about the relative importance of the organizational revitalization dimensions ranged between 3.75 and 3.85. Futurology, with an arithmetic mean of 3.79 and high relative importance, is followed by the dimension of knowledge making with an arithmetic mean of 3.75 and high relative importance. The data of the previous table also indicate that the arithmetic mean of the sample's estimates on the dimensions of the organizational revitalization as



a whole reached 3.80 and high relative importance, meaning that there is a high level of systemic revitalization in the travel and tourism companies operating in Malaysia. The value of the (t) test is 16.49, which is a statistically significant value, as it is greater than the tabular value of 1.96. From the perspective of the investing philosophy side of human forces of Malaysian tourism and travel companies, this study finds that most employees will anticipate the challenges of the upcoming stages and work to find new visions for development in accordance with their companies' contingency plans to deal with R&D and the principle of setting priorities.

**Table 3.** Organizational revitalization dimensions.

| Dimensions | Arithmetic Mean | Standard Deviation | (t) Value | Materiality |
|---|---|---|---|---|
| Work environment | 3.85 | 0.72 | 17.09 | High |
| Knowledge making | 3.75 | 0.78 | 14.04 | High |
| Futurology | 3.79 | 0.79 | 14.48 | High |
| Overall average | 3.80 | 0.70 | 16.49 | High |

Table 4 shows that the average estimates of the sample members regarding the relative importance of the dimensions of entrepreneurial learning ranged between 3.68 and 3.74. The professional experience dimension came first with the highest arithmetic mean (3.74) and high relative importance, followed by the dimension of the transformation process with an arithmetic mean of 3.69 and high relative importance, followed by the dimension of entrepreneurial knowledge, with an arithmetic mean of 3.68 and high relative importance. The data of the previous table also indicate that the arithmetic mean of the estimates of the sample members on the dimensions of entrepreneurial learning as a whole reached 3.70 with high relative importance, meaning that there is a high level of entrepreneurial learning in the travel and tourism companies operating in Malaysia. The value of the (t) test was 13.75, which is a statistically significant value, as it is greater than the tabular value of 1.96.

**Table 4.** Entrepreneurial learning dimensions.

| Dimensions | Arithmetic Mean | Standard Deviation | (t) Value | Materiality |
|---|---|---|---|---|
| Professional experience | 3.74 | 0.82 | 12.99 | High |
| Transformation process | 3.69 | 0.75 | 13.38 | High |
| Entrepreneurial knowledge | 3.68 | 0.81 | 12.16 | High |
| Overall average | 3.70 | 0.74 | 13.75 | high |

*5.3. Hypotheses Testing*

The analysis result of H1 is shown in Figure 2 at the significance level of $\alpha \leq 0.05$. In this study, when the chi-squared per degree of freedom is below six, this shows a reasonable fit, while a ratio between one and two is an excellent fit. The ratio of the model is 2.029, indicating a fairly good fit. The goodness-of-fit index (GFI) is another measure that LISREL provides. The adjusted goodness-of-fit index (AGFI) is an extension of the GFI, adjusted by the ratio of the degrees of freedom for the proposed model to the degrees of freedom for the null model. The GFI for the overall model is 0.92 and the AGFI is 0.90. Other diagnostics for this model include NFI = 0.93, CFI = 0.95, RMR = 0.022, SRMR = 0.018, and RMSEA = 0.031.

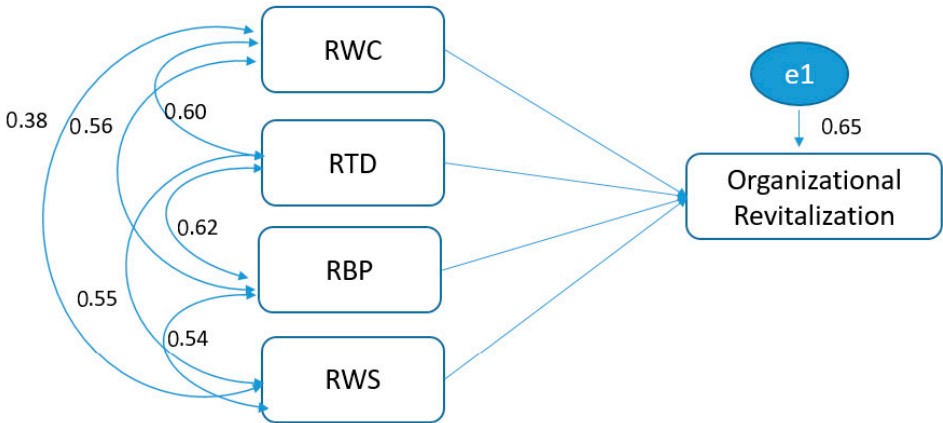

**Figure 2.** The configuration of H1a, H1b, H1c, and H1d.

As shown in Table 5, the value of β in the reengineering work culture dimension reached 0.20, and its T-value is 3.45, with a significance level of Sig = 0.00, indicating that the effect of this dimension in organizational revival is the moral effect. As for the value of β in the reengineering training and development dimension, it reached 0.22 and its T-value is 3.73 with a significance level of Sig = 0.00, indicating that the effect of this dimension in organizational revival is a moral effect. The value of the dimension of reengineering business processes reached 0.32 and its T-value is 4.55, with a significance level of Sig = 0.00. This indicates that the effect of this dimension on organizational revival is a moral effect. The value of the reengineering workplace systems dimension is 0.57, its T-value is 10.84, and the level of significance is Sig = 0.00, which indicates that the effect of this dimension in organizational revitalization is a moral effect. Through the aforementioned values, it is evident that the dimension of reengineering workplace systems as one of the dimensions of reengineering human resources had the greatest impact on the organizational revitalization, followed by the dimension of reengineering work operations, then the dimension of reengineering training and development, and in the last place in terms of impact strength was the reengineering work culture dimension. Based on the results obtained, the first main nihilism hypothesis is rejected and the alternative hypothesis is accepted: there is an effect of reengineering human resources (reengineering work culture, reengineering training and development, reengineering business processes, reengineering workplace systems) on the organizational revitalization of companies in tourism and travel in Malaysia at a significance level of $\alpha \leq 0.05$.

**Table 5.** The analysis of the impact of RHRs on organizational revitalization.

| Hypotheses | β | (t) Value | (Sig) | $R^2$ |
|---|---|---|---|---|
| H1a | 0.20 | 3.45 | 0.00 | |
| H1b | 0.22 | 3.73 | 0.00 | 0.65 |
| H1c | 0.32 | 4.55 | 0.00 | |
| H1d | 0.57 | 10.84 | 0.00 | |

Table 6 presents the results of the hierarchical multiple regression based on two models, as the results of the first model based on the first step reflected the presence of a statistically significant effect of the independent variables represented by reengineering human resources on the organizational revival, where the value of (F) = 124.2, the level of significance is sig F = 0.000, which is less than 0.05, and the value of the determination factor is $R^2$ = 0.650. This indicates that the dimensions of re-engineering human resources have explained 65.0% of the variance in organizational revival. In the second step, a variable (entrepreneurial learning) was introduced to the regression model, where the value of the determination factor $R^2$ increased by 18.2%, and this increase was statistically significant ($\Delta$ F = 184.15) and the level of significance was sig $\Delta$ F = 0.000, which is less than 0.05.

A value of β = 0.679 was found for entrepreneurial learning, with a value of t = 17.869 and a level of significance of Sig = 0.000. The rate of the interpretation of the overall variance improved by 18.2%, increasing from 65.0% to 83.2%. Therefore, this rejects the second nihilistic main hypothesis and accepts the alternative hypothesis, which states: Entrepreneurial learning modifies the effect of reengineering human resources on the organizational revival of travel and tourism companies in Malaysia at the significance level of $\alpha \leq 0.05$.

**Table 6.** Multiple hierarchical regression analysis of modifying RHRs.

| Dependent Variable | Independent Variables | The First Model | | | The Second Model | | |
|---|---|---|---|---|---|---|---|
| | | β | (t) Value | Sig. | β | (t) Value | Sig t |
| Reengineering work culture | Organizational revitalization | 0.20 | 3.45 | 0.00 | 0.603 | 15.801 | 0.000 |
| | Reengineering training and development | 0.22 | 3.73 | 0.00 | 0.331 | 5.601 | 0.000 |
| | Reengineering business processes | 0.32 | 4.55 | 0.00 | 0.413 | 7.88 | 0.030 |
| | Reengineering workplace systems | 0.57 | 10.84 | 0.00 | 0.64 | 16.98 | 0.029 |
| | Entrepreneurial learning | | | | 0.679 | 17.869 | 0.000 |
| | The coefficient of determination ($R^2$) | | 0.650 | | | 0.832 | |
| | $\Delta R^2$ | | 0.650 | | | 0.182 | |
| | $\Delta F$ | | 124.21 | | | 184.15 | |
| | Sig $\Delta F$ | | 0.000 | | | 0.000 | |

## 6. Conclusions

### 6.1. Discussions

The results related to H1 show that there is a statistically significant effect for all dimensions of reengineering human resources on the organizational revitalization of tourism and travel companies in Malaysia, where the determination factor is 0.65, the regression values increase for all dimensions, and the level of statistical significance is less than 0.05. This indicates that tourism and travel companies seek to apply reengineering human resources in the dimensions represented in reengineering (work culture, training and development, business processes, workplace systems) and that the application of reengineering human resource processes positively affects and increases organizational revitalization processes. The results of the hypotheses branching from the main hypothesis showed the existence of an impact of RHRs (reengineering work culture, reengineering training and development, reengineering work processes, reengineering workplace systems) on the work environment, knowledge-making, and futurology separately in Malaysian travel and tourism companies.

This study attributes this result to the fact that the application of RHR dimensions by tourism and travel companies increases the organizational adaptation in the work environment and is characterized by comfort, quality and flexibility, and the ability to increase knowledge-making in terms of support and motivation for workers in possessing and sharing information and applying the resulting knowledge and skills in the performance of businesses and achieving goals. The results of the current study are in agreement with [15], whose results showed the existence of a statistically significant effect of reengineering human resources dimensions in strengthening high-performance work systems in economic institutions, the sample of the study, and are in agreement with the study of [44], which indicates the effect of RHRs on the performance of employees in Jordanian telecommunications companies.

The results of the study show a statistically significant effect at the level $0.05 \geq \alpha$ of pioneering/entrepreneurial learning in modifying (improving) the effect of reengineering human resources in the organizational revitalization of tourism and travel companies in Malaysia. The change in the value of the explanatory power of the model amounts to $\Delta R^2 = 0.182$, which means that the moderating variable (entrepreneurial learning) explains an amount (18.2%) of increase and clarifies the effect of the independent variable (reengineering human resources) on the dependent variable (organizational revitalization) to raise the value of variance in the interpretation of the overall model from 0.650 to 0.832.

This study explains this result from improving and increasing the impact on the travel and tourism companies' application of entrepreneurial learning effectively, which leads to an increase and improvement in the impact of RHRs on the dimensions of organizational revitalization so that the application of entrepreneurial learning in travel and tourism companies works to provide workers with the information, data, knowledge, experiences, and skills necessary to improve performance and identify risks and threats facing companies and make much more effort to solve them. Consequently, RHRs maintain the continuity of companies' work within the business environment and increase the organization's competitiveness and capability to achieve goals.

*6.2. Implications and Suggestions*

This study puts forward a set of implications and suggestions based on the findings, as follows: (1) The maintenance of travel and tourism companies on continuing interest in RHRs and focusing on reengineering work culture through developing work values, developing innovative solutions to solve problems, applying empowerment practices, and enhancing skills that add value to performance. (2) Travel and tourism companies focus on RHRs by applying training and development engineering, working on identifying training needs, developing training programs that rely on diversity in skills, and simulating global competitive forces, providing workers with new skills and refining their creativity. (3) Urging tourism and travel companies to pay attention to entrepreneurial learning by investing professional expertise, converting experiences into knowledge, investing available knowledge, supporting opinions based on previous experiences, and enhancing the role of expertise in creativity among workers. (4) Creating a work environment in tourism and travel companies to support the skills of their workers by adopting entrepreneurial learning and innovative ideas. (5) Seeking to identify the different training needs of workers in travel and tourism companies and to rely on the training programs appropriate to the nature of business in travel and tourism companies along with future competition trends.

**Author Contributions:** Conceptualization, methodology, and supervision, C.-W.L.; Data curation and investigation, C.L.P.; Visualization and formal analysis, H.C.C. All authors have read and agreed to the published version of the manuscript.

**Funding:** This research received no external funding.

**Informed Consent Statement:** Informed consent was obtained from all subjects involved in the study.

**Conflicts of Interest:** The authors declare no conflict of interest.

## Appendix A. Questionnaire

Dear all,

The purpose of this questionnaire is to understand how much effect of engineering human resources and entrepreneurship learning on organizational revitalization.

Please fill in the answers carefully and truthfully based on your actual situations. All information obtained in this questionnaire is only used as a reference for academic research, thank you!

Sincerely yours,

Prof. Ph.D. Cheng-Wen Lee,

Chung Yuan Christian University, Taiwan

No. 200, Zhongbei Rd., Zhongli Dist., Taoyuan City 320314, Taiwan (R.O.C.)

E-mail: chengwen@edu.edu.tw

Risks and Benefits: This questionnaire is conducted anonymously and my participation in this research does not involve any significant risks. In terms of income, it is beneficial to research related to service-learning through the questionnaire.

Voluntary Participation: This questionnaire is my voluntary participation, and I can interrupt filling out the questionnaire at any time.

Statement of Consent: Once this questionnaire is clicked to start, it means that I have agreed, which is regarded as a signed written consent. I agree that the researcher of this study may use my records to provide for current and future research.

1.  Basic information:

    Gender: □male □female;

    Age: □30 years or less □31-less than 40 years □41-less than 50 years
    □51-less than 60 years □60 years and over

    Educational level: □High school or community diploma □Bachelor's degree
    □Higher diploma □Master's degree □Doctoral degree

    Work position: □Director □Deputy General Manager □Head of the Department
    □Employee

    Years of experience: □5 years or less □6~10 years □11~15 years □16 years and over

2.  Engineering Human Resources Survey. Please select it if you feel it is appropriate.

    Number 5 means "Strongly Agree", 4 is "Agree", 3 is "Neutral", 2 is "Disagree", and 1 is "Strongly Disagree".

|   |   | 5 | 4 | 3 | 2 | 1 |
|---|---|---|---|---|---|---|
| 1 | To what extent do you consider that reshaping the personality of organization plays an important role in your job career? | □ | □ | □ | □ | □ |
| 2 | To what extent do you consider that the embodied organization's mission and values in the common understanding plays an important role in your job decision-making? | □ | □ | □ | □ | □ |
| 3 | To what extent do you consider that the relationship of subordinates with your managers and vice versa can enhance the sense of membership? | □ | □ | □ | □ | □ |
| 4 | To what extent do you consider that activities at all levels, and belonging/exchange of ideas within the organization can enhance the organization's sustainability? | □ | □ | □ | □ | □ |
| 5 | To what extent do you consider that the process of improving and adding value to incomplete human resources can perform new roles? | □ | □ | □ | □ | □ |
| 6 | To what extent do you consider that learning is provided to employees while they are on the job can enhance their levels of knowledge and skills? | □ | □ | □ | □ | □ |

| | | 5 | 4 | 3 | 2 | 1 |
|---|---|---|---|---|---|---|
| 7 | To what extent do you consider that the organization's task and its responsibilities are shared by both training and development for direct managers? | ☐ | ☐ | ☐ | ☐ | ☐ |
| 8 | To what extent do you consider that a radical reconstruction and restructuring of processes can achieve organization's fundamental improvements? | ☐ | ☐ | ☐ | ☐ | ☐ |
| 9 | To what extent do you consider that critical areas for economic growth and the main motivation can ensure the development of the organization's structure to face crisis? | ☐ | ☐ | ☐ | ☐ | ☐ |
| 10 | To what extent do you consider that composition and structure of jobs-duration-cost-size of products or services-quality in connection with reengineering can change the basic principles of organization? | ☐ | ☐ | ☐ | ☐ | ☐ |
| 11 | To what extent do you consider that re-developing and organizing the workplace is a crucial organizational approach? | ☐ | ☐ | ☐ | ☐ | ☐ |
| 12 | To what extent do you consider that through a system using visual cues and visual management to organize the workplace can help reduce some forms of waste and show problems sooner? | ☐ | ☐ | ☐ | ☐ | ☐ |
| 13 | To what extent do you consider that each element being in the right place and position of WMS/5S can enhance the operational efficiency? | ☐ | ☐ | ☐ | ☐ | ☐ |
| 14 | To what extent do you consider that a set of performance and special knowledge and skills acquired during practice or through experience can achieve a high level of performance, and reduce expenses? | ☐ | ☐ | ☐ | ☐ | ☐ |
| 15 | To what extent do you consider that skills, competencies and innate observation strength can achieve a high level of performance and reduce expenses? | ☐ | ☐ | ☐ | ☐ | ☐ |
| 16 | To what extent do you consider that entrepreneurial learning, an experiential process through which personal experience in entrepreneurship can be transformed into knowledge? | ☐ | ☐ | ☐ | ☐ | ☐ |
| 17 | To what extent do you consider that a group of activities can create new outputs and add value, and in turn can be used to guide the selection of new experiences? | ☐ | ☐ | ☐ | ☐ | ☐ |
| 18 | To what extent do you consider that information and experiences of entrepreneurs gain about contacts and relationships with people and stakeholders can involve the company's business? | ☐ | ☐ | ☐ | ☐ | ☐ |
| 19 | To what extent do you consider that potential new viable markets, product availability, resources and competitive response can enhance the ability to respond to change and seize/discover entrepreneurship opportunities? | ☐ | ☐ | ☐ | ☐ | ☐ |

| | | 5 | 4 | 3 | 2 | 1 |
|---|---|---|---|---|---|---|
| 20 | To what extent do you consider that dealing with obstacles, problems and technological developments in different ways can achieve greater success? | ☐ | ☐ | ☐ | ☐ | ☐ |
| 21 | To what extent do you consider that every part of the employee's participation in the work itself, such as relationships with colleagues and bosses can be connected with organizational culture? | ☐ | ☐ | ☐ | ☐ | ☐ |
| 22 | To what extent do you consider that all the physical conditions around the workplace can affect employees in a way? | ☐ | ☐ | ☐ | ☐ | ☐ |
| 23 | To what extent do you consider that direct or indirect, physical factors include workplace air temperature, work area, noise, density, and area can form working atmosphere? | ☐ | ☐ | ☐ | ☐ | ☐ |
| 24 | To what extent do you consider that non-material work environment such as the relationship between colleagues, bosses and subordinates can be felt by employees? | ☐ | ☐ | ☐ | ☐ | ☐ |
| 25 | To what extent do you consider that values and trends prevailing in administrative companies, a healthy work climate can improve productivity? | ☐ | ☐ | ☐ | ☐ | ☐ |
| 26 | To what extent do you consider that maximizing intellectual capital/information and further measuring it in modern technological ways can create new value? | ☐ | ☐ | ☐ | ☐ | ☐ |
| 27 | To what extent do you consider that through improving efficiency and effectiveness among individuals and cooperation can enhance innovation and decision-making? | ☐ | ☐ | ☐ | ☐ | ☐ |
| 28 | To what extent do you consider that outputs used to gain market share by arranging the accumulated ideas and experiences using advanced technology can achieve a competitive advantage? | ☐ | ☐ | ☐ | ☐ | ☐ |
| 29 | To what extent do you consider that an organized scientific effort of long-term planning, forecasting and anticipation of future developments can produce rapid developments? | ☐ | ☐ | ☐ | ☐ | ☐ |
| 30 | To what extent do you consider that to exploit opportunities, avoid threats, and find alternatives/solutions can reach an advanced stage of knowledge of future trends? | ☐ | ☐ | ☐ | ☐ | ☐ |

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
