# Peer review of "Reengineering Human Resources and Entrepreneurial Learning towards Organizational Revitalization in Malaysian Travel and Tourism Companies during the COVID-19 Pandemic"

_sustainability, doi:10.3390/su142013509_

Round 1
Reviewer 1 Report
Dear Authors,
After reading the first version of your manuscript, I am still not fully persuaded that the article goes into detail enough to provide added value to Sustainability readership.
This time, I will try to give my comments in a more clear version:
The author needs to clarify the new contribution of the research in the introduction. It is necessary to clearly state the new and motivating points of the article.
The literature review should be placed after the missing section. Authors need to update recent studies. And point out the missing point to carry out this study. The author should have a literature review to compare the results of previous studies conducted in the same research context.
The author needs to make a clear hypothesis for each pair of moderator variables.
The author needs to fully evaluate the indicators for each scale and the whole model, for example, CFI, RMSEA,...CMIN/df,...
The author needs to create an interaction variable to test the moderator hypothesis.
I hope my comments may help you in developing the paper.
Author Response
Point 1: After reading the first version of your manuscript, I am still not fully persuaded that the article goes into detail enough to provide added value to Sustainability readership. This time, I will try to give my comments in a more clear version: The author needs to clarify the new contribution of the research in the introduction. It is necessary to clearly state the new and motivating points of the article.
Response 1: Add several paragraphs in the Introduction, for instant,
“The understanding of influential human forces in current market supports the formation of human resources strategy. The transition to reengineering and reconfiguration of human resources provides a clear roadmap for utilizing human capital in reviving both organization’s goals and challenges [3]. However, organizational revitalization is a complex process that requires many changes, capabilities and solutions, needs to be implemented and accepted by the institution and stakeholders, and eventually brings the organizational value, vitality and enthusiasm [3]. An organization puts a lot of effort into adapting internal and external pressures on reengineering human resources.”
“In light of the mentioned above, the study aims to test the relationship of the three variables, including the impact of reengineering human resources on organizational revitalization, and the existence of pioneering/entrepreneurial learning, as a moderating variable. We use the tourism and travel companies in Malaysia as our study sample, because these companies are the most in need at present for organizational revitalization due to being affected by repercussions of the COVID 19 crisis.”
Point 2: The literature review should be placed after the missing section. Authors need to update recent studies. And point out the missing point to carry out this study. The author should have a literature review to compare the results of previous studies conducted in the same research context. The author needs to make a clear hypothesis for each pair of moderator variables. The author needs to fully evaluate the indicators for each scale and the whole model, for example, CFI, RMSEA,...CMIN/df,.... The author needs to create an interaction variable to test the moderator hypothesis.
Response 2: Add several paragraphs, for instant,
“The present study is a causal study, with a quantitatively through questionnaires (see Appendix). It adopts the descriptive analytical approach method that is associated with describing the results with their analysis. The field of study consists of travel and tourism companies in Malaysia, which are registered with the Ministry of Tourism, Arts and Culture (2021), whose financial guarantee amounts to USD25,000, and the number is 63 companies, and it contains the three administrative levels -- higher, central and supervisory, as for the study population; it may consist of all workers in the tourism and travel companies, within the aforementioned levels (upper, middle and supervisory management), whose number is 768 employees of Malaysian Association of Tour and Travel Agents. The result is 239 valid responses (31.1%).”
2
“To evaluate the relative importance of different question items, each respondent specifies the relative importance of characteristics of RHRs, entrepreneurship learning, and organizational revitalization on a 5-point Likert scale, indicating whether relative importance is very low (1) or very high (5). In this study, the method of reliability measurement is Cronbach’s coefficient α. Cronbach’s coefficient α is calculated for each of these factors to assess the internal consistency of the model constructs. According to Price and Mueller [43], a standard coefficient α of 0.60 or higher generally is considered acceptable when using a measure. If statistical significance is not achieved, the research may need to eliminate the indicator or attempt to transform it for a better fit with the construct. Here, the values of Cronbach’s coefficient α of all variables exceed 0.80. This indicates that the research has good consistency and stability.”
“The analysis result of H1 is shown in Figure 2 at the significance level (α ≤ 0.05). To test these hypotheses, we use the Structured Equation Model (SEM) method through the (AMOS) program. In this study, when the chi-square per degree of freedom is below six, this shows a reasonable fit, while a ratio between one and two is an excellent fit. The ratio of the model is 2.029, indicating a fairly good fit. The model fit assessment approach is involved, using several diagnostics to judge the simultaneous fit of the measurement and structural models to data collected for this study. The goodness-of-fit index (GFI) is another measure LISREL provides. The adjusted goodness-of-fit index (AGFI) is an extension of the GFI, adjusted by the ratio of degrees of freedom for the proposed model to the degrees of freedom for the null model. The GFI for the overall model is 0.92 and the AGFI is 0.90. Other diagnostics for this model include NFI=0.93, CFI=0.95, RMR=0.022, SRMR=0.018, and RMSEA=0.031.”

Reviewer 2 Report
The article structure and content are good, but it shows 20 percent similarity ask the author to rewrite.
Add one para describing the source of the questionnaire / instrument used in the study
Author Response
Point 1: The article structure and content are good, but it shows 20 percent similarity ask the author to rewrite.
Response 1: Already reviewed, corrected, revised and rewrote many places. Please refer the red word in this article. The similarity already is down to 12%.
Point 2: Add one para describing the source of the questionnaire / instrument used in the study.
Response 2: Add several paragraphs, for instant,
“The present study is a causal study, with a quantitatively through questionnaires (see Appendix). It adopts the descriptive analytical approach method that is associated with describing the results with their analysis. The field of study consists of travel and tourism companies in Malaysia, which are registered with the Ministry of Tourism, Arts and Culture (2021), whose financial guarantee amounts to USD25,000, and the number is 63 companies, and it contains the three administrative levels -- higher, central and supervisory, as for the study population; it may consist of all workers in the tourism and travel companies, within the aforementioned levels (upper, middle and supervisory management), whose number is 768 employees of Malaysian Association of Tour and Travel Agents. The result is 239 valid responses (31.1%).”
“To evaluate the relative importance of different question items, each respondent specifies the relative importance of characteristics of RHRs, entrepreneurship learning, and organizational revitalization on a 5-point Likert scale, indicating whether relative importance is very low (1) or very high (5). In this study, the method of reliability measurement is Cronbach’s coefficient α. Cronbach’s coefficient α is calculated for each of these factors to assess the internal consistency of the model constructs. According to Price and Mueller [43], a standard coefficient α of 0.60 or higher generally is considered acceptable when using a measure. If statistical significance is not achieved, the research may need to eliminate the indicator or attempt to transform it for a better fit with the construct. Here, the values of Cronbach’s coefficient α of all variables exceed 0.80. This indicates that the research has good consistency and stability.”
“The analysis result of H1 is shown in Figure 2 at the significance level (α ≤ 0.05). To test these hypotheses, we use the Structured Equation Model (SEM) method through the (AMOS) program. In this study, when the chi-square per degree of freedom is below six, this shows a reasonable fit, while a ratio between one and two is an excellent fit. The ratio of the model is 2.029, indicating a fairly good fit. The model fit assessment approach is involved, using several diagnostics to judge the simultaneous fit of the measurement and structural models to data collected for this study. The goodness-of-fit index (GFI) is another measure LISREL provides. The adjusted goodness-of-fit index (AGFI) is an extension of the GFI, adjusted by the ratio of degrees of freedom for the proposed model to the degrees of freedom for the null model. The GFI for the overall model is 0.92 and the AGFI is 0.90. Other diagnostics for this model include NFI=0.93, CFI=0.95, RMR=0.022, SRMR=0.018, and RMSEA=0.031.”

Reviewer 3 Report
Dear authors, I think the idea of your research is very interesting. However, the document must be improved so much to have a better understanding. I hope the quality of your work can improve by attending the following comments of my feedback:
Keywords: I think that words such as Reengineering Human Resources; Entrepreneurial Learning; Organizational Revitalization; Corporate Sustainability must be in the keywords.
Line 38: In the sentence “…bring the disadvantages…”, change the word “bring” by “brings” and delete the word “the”.
Line 39: In the sentence “However, other sectors that also…”, delete the word “that”.
Line 42: What career do you mean to?
Line 51: Define the acronym GDP.
Lines 51-52: The citation International Labor Organization (2000) must be in numerical format, for instance: “International Labor Organization (ILO) [3] reveals….”
Lines 65-73: The following sentence is confusing: “As the understanding of human resources of the current market conditions and influencing forces, will support the formation of the required human resources strategy, and the transition to reengineering and reconfiguration of human resources in order to provide a clear roadmap for utilizing the human capital of the organization in addressing both goals and challenges, and reviving the organization [3]”.
Please, review and correct it. I suggest doing short sentences to improve their understanding.
Also, the following sentences in confusing, you repeat the words “organization/organizational”, “process”, and “change” in a relatively short space:
However, organizational revitalization is a complex process that requires many changes and capabilities as a process of organizational change that is implemented and accepted by the organization, and brings value, vitality and enthusiasm [3].
The same with the following sentence:
“An organization to adapt to internal and external pressures”.
Line 73: You mention “It is a very powerful global formula for changing the culture,….”, what is a very powerful global formula? Please, specify, it is not clear.
Line 76: In the sentence “…the importance part of innovation,…”, change the word “importance” to “important”
Line 81: The following sentence is confusing, please, review and correct it: “…as it is the most in need at the present time for the organizational revival as it is one of the most companies have been…”
Line 86: In the sentence “The experience of organizations suggest…”, change the word “suggest” to “suggests” since “experience” is singular.
Line 87: Define the acronym HHR.
Lines 86-90: The following sentence is confusing: “The experience of organizations suggest that if the new relationship between personnel and line managers has to work during initial phases of HRR, and then the line managers should be considered efficient enough to take on greater responsibilities related to personnel management activities”.
You mention the word “if” and also you mention the words “and then” (it is understood as a second if). If you remove the word “and”, then the sentence will be as follows: “The experience of organizations suggest that if the new relationship between personnel and line managers has to work during initial phases of HRR, then the line managers should be considered efficient enough to take on greater responsibilities related to personnel management activities”.
In this last sentence, there is an “if” (cause) and there is a “then” (consequence).
Lines 96-100: The following sentence is confusing: “All large companies have gained access to the same mental strength of strategy, technology and capital, so the resources have become human and organizational capacity that can be mobilized is the only source of a company’s unique competitive advantage, to do everything better than competitors’ achievement, and to quickly exploit fewer organization’s scarce resources [7]”.
Review and correct it.
Line 105: In the sentence “The most important of which is…”, what do you refer with “which”? The most important of what? Problems? Processes? Tools? Methods? It is not clear, please specify.
Line 105: In the sentence “The most important of which is the restoration human resource engineering in the organization”, add the word “of” before human.
Lines 109-114: The following two sentences are confusing: “Reengineering human resource as a process aimed at transforming human resource skills, capabilities and knowledge into intellectual capital and employing it in the general goals of the organization and the relevant parties [9]. The complete review of all critical human resources measures that contribute to raising the efficiency of the organization’s work and enhancing its ability to provide higher quality services with greater effectiveness [10]”.
Line 127: In the sentence “[13, 14] encourage and support the transition”, write the names of the authors as follows: “Christenson [13] and Boudreau and Ramstad [14] encourage and support the transition…”
Lines 128-130: The following sentence is confusing: “The extent of the contribution of RHR in providing and strengthening high performance systems or practices, by surveying the opinions of managers in some economic institutions in Iran [15]”.
Review and correct it.
Lines 130-132: As it is redacted, this seems to be methodology: “A questionnaire survey as a study tool is distributed to workers at higher administrative levels in seven service and industrial economic institutions in Iran”.
Lines 132: The most important of what in the following sentence? “The most important of which is the…”
The most important institution? The most important level? It is not clear. Please, specify.
Line 136: A sudden change of subject is felt, you start talking about the attitudes of the students. As a reader, one wonders which students? You have to go step by step, for example: “In another research, Smith [16] analyzed the attitudes of university students .....”
Section 2.2: In section 2.2. Reengineering Human Resources, you talk so much about Entrepreneurial learning. This causes confusion, it seems this section is dedicated more to s entrepreneurial learning than to Reengineering Human Resources.
In general, section 2. Literature review is difficult to read and understand, some ideas do not make sense, the topic is abruptly changed.
Lines 160-166: The following sentences are confusing since you talk in the present and past tense at the same time: “Entrepreneurial learning is a dynamic process that occurs through the interaction between the personal and social experience of the entrepreneur during entrepreneurial journey; that is independent of the unique context of the creative industry, and the learning procedures for creative entrepreneurs were characterized by personal experiences coupled with a high level of cooperation and participation. In short, entrepreneurial learning affected the entrepreneurial efficiency of women entrepreneurs [18]”.
Lines 264-265: The following sentence is confusing: “Global, to continue growth and achieve profits or maintain stability, and at a minimum, stop losses and maintain survival”.
Review and correct it.
Line 263: You mention “…clutches of tempering in the economic system”, but tempering has no clutches. Be more technical.
Line 280: In the sentence “And to promote the process of systemic revitalization of certain divisions [34]”, write “and” instead of “And”.
Section 3. Research Design: In this section, you do not talk anything about Research Design, it seems literature review.
Lines 286-287: Here there is an abrupt change of subject. You introduce the hypotheses abruptly, you must make a connection between paragraphs, and between the ideas you intend to give.
Figure 1: All figures and tables (including those in the appendix) must be mentioned in the text. For instance: Figure 1 shows the hypothetical causal model.
Section 3.2 Variables Definitions: The variables must be defined before presenting the hypotheses. Again, I suggest giving a definition of each variable in section 2. Literature review. Once you give a definition, you can discuss previous research on the relationship between the variables, and finally, you can propose the hypothesis.
I suggest reading the following articles and use them as a guide:
https://www.mdpi.com/2227-7390/9/9/971/htm
https://www.mdpi.com/2071-1050/12/21/9194/htm
4. Methodology: The methodology should answer the question "How is the research carried out to achieve the objectives that were set? It should be described in as much detail as possible, mentioning what materials or instruments are used (e.g., software, measuring instruments, etc.). In this section, you are mentioning Results, for example, the results of the sample distribution and the results of Reengineering Human Resources Dimensions. The results should be presented in a separate section called Results. Mixing methodology and results makes it difficult to read and understand the manuscript.
Be sure to validate the questionnaire.
General comments
1. Some sentences need citations, for instance, the following sentences need references. Review in all the manuscript.
In addition to the COVID-19 pandemic, the issue of travel insurance has come to the fore, which may be a way to revive this sector by providing travel packages with travel insurance services that may reduce the concerns that threaten this sector. Given the high level of the importance of tourism as a contributor to GDP and a major source of income and jobs, this pandemic affects economies and labor markets due to international travel restrictions and procedures. The closures imposed by governments in many countries and the recommendations related to staying at home also deal a blow to domestic tourism, which will lead to a contraction and stunt growth in the economies of many countries, especially Asian ones.
2. I suggest including, in section 2. Literature review, subsections that talk about the background of the impact of one variable on another variable. For example, include a subsection called 2.1 Impact of Reengineering of Human Resources on Organizational Revitalization, cite research about the relation of these two variables, and so on with the other variables.
3. In general, the writing must be improved. It is difficult to read and understand the document. English must be improved.
4. The title should be changed to something that makes more reference to the objective of the research. Take the articles I shared above to define a better title.
Best wishes
Author Response
Point 1:
Keywords: I think that words such as Reengineering Human Resources; Entrepreneurial Learning; Organizational Revitalization; Corporate Sustainability must be in the keywords.
Line 38: In the sentence “…bring the disadvantages…”, change the word“bring” by “brings” and delete the word “the”.
Line 39: In the sentence “However, other sectors that also…”, delete the word “that”.
Line 42: What career do you mean to?
Line 51: Define the acronym GDP.
Lines 51-52: The citation International Labor Organization (2000) must be in numerical format, for instance: “International Labor Organization (ILO) [3] reveals….”
Lines 65-73: The following sentence is confusing: “As the understanding of human resources of the current market conditions and influencing forces, will support the formation of the required human resources strategy, and the transition to reengineering and reconfiguration of human resources in order to provide a clear roadmap for utilizing the human capital of the organization in addressing both goals and challenges, and reviving the organization [3]”
Please, review and correct it. I suggest doing short sentences to improve their understanding.
Response 1: Added the three terms in keywords. According to the reviewer’s direction, we already correct the grammar mistakes and shorten the length of sentences. i.e. “The understanding of influential human forces in current market supports the formation of human resources strategy. The transition to reengineering and reconfiguration of human resources provides a clear roadmap for utilizing human capital in reviving both organization’s goals and challenges [3].”
Point 2: Also, the following sentences in confusing, you repeat the words“organization/organizational”, “process”, and “change” in a relatively short space:
However, organizational revitalization is a complex process that requires many changes and capabilities as a process of organizational change that is implemented and accepted by the organization, and brings value, vitality and enthusiasm [3]. The same with the following sentence: “An organization to adapt to internal and external pressures”.
Response 2: We rewrote the sentences as follows. “However, organizational revitalization is a complex process that requires many changes, capabilities and solutions, needs to be implemented and accepted by the institution and stakeholders, and eventually brings the organizational value, vitality and enthusiasm [3].” An organization puts a lot of effort into adapting internal and external pressures on reengineering human resources. “
Point 3:
Line 73: You mention “It is a very powerful global formula for changing the culture,….”, what is a very powerful global formula? Please, specify, it is not clear.
Line 76: In the sentence “…the importance part of innovation,…”, hange the word “importance” to “important”
Line 81: The following sentence is confusing, please, review and correct it: “…as it is the most in need at the present time for the organizational revival as it is one of the most companies have been…”
Line 86: In the sentence “The experience of organizations suggest…”, change the word “suggest” to “suggests” since “experience” is singular.
Line 87: Define the acronym HHR.
Response 3: Corrected the grammar errors and rewrote the confused sentences as follows. “In light of the mentioned above, the study aims to test the relationship of the three variables, including the impact of reengineering human resources on organizational revitalization, and the existence of pioneering/entrepreneurial learning, as a moderating variable. We use the tourism and travel companies in Malaysia as our study sample, because these companies are the most in need at present for organizational revitalization due to being affected by repercussions of the COVID 19 crisis.
Point 4:
Lines 86-90: The following sentence is confusing: “The experience of organizations suggest that if the new relationship between personnel and line managers has to work duringnitial phases of HRR, and then the line managers should be considered efficient enough to take on greater responsibilities related to personnel management activities”.
You mention the word “if” and also you mention the words “and then” (it is understood as a second if). If you remove the word “and”, then the sentence will be as follows: “The experience of organizations suggest that if the new relationship between personnel and line managers has to work during initial phases of HRR, then the line managers should be considered efficient enough to take on greater responsibilities related to personnel management activities”.
In this last sentence, there is an “if” (cause) and there is a “then” (consequence).
Response 4: Deleted “if”
Point 5:
Lines 96-100: The following sentence is confusing: “All large companies have gained access to the same mental strength of strategy, technology and capital, so the resources have become human and organizational capacity that can be mobilized is the only source of a company’s unique competitive advantage, to do everything better than competitors’ achievement, and to quickly exploit fewer organization’s scarce resources [7]”.
Review and correct it.
Response 5: Rewrote to “All large companies have gained access to rare and unreplaceable resources of strategic strength, technology and capital, which have become human and organizational capacity that can be mobilized to the company’s unique competitive advantage better than competitors’ achievement [7]”.
Point 6:
Line 105: In the sentence “The most important of which is…”, what do you refer with “which”? The most important of what? Problems? Processes? Tools? Methods? It is not clear, please specify.
Line 105: In the sentence “The most important of which is the restoration human resource engineering in the organization”, add the word “of” before human.
Response 6: Deleted “of which”.
Point 7:
Lines 109-114: The following two sentences are confusing: “Reengineering human resource as a process aimed at transforming human resource skills, capabilities and knowledge into intellectual capital and employing it in the general goals of the organization and the relevant parties [9]. The complete review of all critical human resources measures that contribute to raising the efficiency of the organization’s work and enhancing its ability to provide higher quality services with greater effectiveness [10]”.
Response 7: Rewrote to “RHRs are transforming human resource skills, capabilities and knowledge into intellectual capital and achieving the organization’s goals and relevant parties [9]. The comprehensive review of critical human resources contributes to raising the efficiency of the organization’s work and to enhancing its ability and higher quality services [10].”
Point 8: Line 127: In the sentence “[13, 14] encourage and support the transition”, write the names of the authors as follows: “Christenson [13] and Boudreau and Ramstad [14] encourage and support the transition…”.
Response 8: Already corrected.
Point 9:
Lines 128-130: The following sentence is confusing: “The extent of the contribution of RHR in providing and strengthening high performance systems or practices, by surveying the opinions of managers in some economic institutions in Iran [15]”.
Lines 130-132: As it is redacted, this seems to be methodology: “A questionnaire survey as a study tool is distributed to workers at higher administrative levels in seven service and industrial economic institutions in Iran”.
Lines 132: The most important of what in the following sentence? “The most important of which is the…”.
The most important institution? The most important level? It is not clear. Please, specify.
Review and correct it. Response 9: Rewrote to “An empirical result finds that the contribution of RHRs provides and strengthens high performance systems or practices, on the basis of surveying on the opinions of managers in some economic institutions in Iran [15].”
Point 10:
Lines 160-166: The following sentences are confusing since you talk in the present and past tense at the same time: “Entrepreneurial learning is a dynamic process that occurs through the interaction between the personal and social experience of the entrepreneur during entrepreneurial journey; that is independent of the unique context of the creative industry, and the learning procedures for creative entrepreneurs were characterized by personal experiences coupled with a high level of cooperation and participation. In short, entrepreneurial learning affected the entrepreneurial efficiency of women entrepreneurs [18]’.
Response 10: We all revised to the present tense in this article.
Point 11:
Lines 264-265: The following sentence is confusing: “Global, to continue growth and achieve profits or maintain stability, and at a minimum, stop losses and maintain survival’.
Line 263: You mention “…clutches of tempering in the economic system”, but tempering has no clutches. Be more technical.
Line 280: In the sentence “And to promote the process of systemic revitalization of certain divisions [34]”, write “and” instead of “And”
Review and correct it.
Response 11: Already corrrected the grammar errors. Revised to “To continue growth, achieve profits or maintain stability, at a minimum stop losses and maintain survival are very crucial in global economy.”
Point 12:
Section 3. Research Design: In this section, you do not talk anything about Research Design, it seems literature review.
Lines 286-287: Here there is an abrupt change of subject. You introduce the hypotheses abruptly, you must make a connection between paragraphs, and between the ideas you intend to give.
Figure 1: All figures and tables (including those in the appendix) must be mentioned in the text. For instance: Figure 1 shows the hypothetical causal model. Section 3.2 Variables Definitions: The variables must be defined before presenting the hypotheses. Again, I suggest giving a definition of each variable in section 2. Literature review. Once you give a definition, you can discuss previous research on the relationship between the variables, and finally, you can propose the hypothesis.
Response 12: We change the sub-title to “3. Research Conceptual Framework”. Already put some literature to make a connection between varibales and revised the whole Section 3.
Point 13: Methodology: The methodology should answer the question "How is the research carried out to achieve the objectives that were set? It should be described in as much detail as possible, mentioning what materials or instruments are used (e.g., software, measuring instruments, etc.). In this section, you are mentioning Results, for example, the results of the sample distribution and the results of Reengineering Human Resources Dimensions. The results should be presented in a separate section called Results. Mixing methodology and results makes it difficult to read and understand the manuscript. Be sure to validate the questionnaire.
Response 13: According to the reviewer’s suggestion, we add the explanation “To evaluate the relative importance of different question items, each respondent specifies the relative importance of characteristics of RHRs, entrepreneurship learning, and organizational revitalization on a 5-point Likert scale, indicating whether relative importance is very low (1) or very high (5). In this study, the method of reliability measurement is Cronbach’s coefficient α. Cronbach’s coefficient α is calculated for each of these factors to assess the internal consistency of the model constructs. According to Price and Mueller [43], a standard coefficient α of 0.60 or higher generally is considered acceptable when using a measure. If statistical significance is not achieved, the research may need to eliminate the indicator or attempt to transform it for a better fit with the construct. Here, the values of Cronbach’s coefficient α of all variables exceed 0.80. This indicates that the research has good consistency and stability.”
Point 14: In general, the writing must be improved. It is difficult to read and understand the document. English must be improved.
Response 14: The grammatical errors and writing style in the original paper have been corrected by a professor who is a native English speaker. I believe this revised version will much better than the original copy in English writing.
Point 15: The title should be changed to something that makes more reference to the objective of the research. Take the articles I shared above to define a better title.
https://www.mdpi.com/2071-1050/12/21/9194/htm
https://www.mdpi.com/2227-7390/9/9/971/htm
Response 15: We change the paper title to
“Reengineering Human Resources and Entrepreneurial Learning towards Organizational Revitalization during the COVID-19 Pandemic in Malaysian Travel and Tourism Companies

Round 2
Reviewer 1 Report
Congratulations team.
Author Response
Please refer the attachment. Thank you very much for your time in reviewing our paper.

Reviewer 3 Report
Dear authors, the topic of your research is interesting, and the manuscript has been improved. However, it requires more improvements. Take into account the following comments:
-The methodology must be improved. It is very superficial. It must be described in more detail.
-Information presented in Tables 1-6 does not correspond to Methodology, it corresponds to Results. In Methodology, you have to explain HOW you did your research to obtain those Results.
- Basically, in Methodology, you must explain how you perform the research to obtain the characteristics of respondents, the descriptive statistics, and how you performed the hypothesis testing. Then, in the Results section, you must show the Results you obtained after you applied the Methodology.
- Include a section of 5. Results, and its corresponding subsections 5.1 Characteristic of Respondents, 5.2 Descriptive Statistics, 5.3 Hypothesis Testing
-English was not improved.
I hope my comments help improve your document.
Best wishes.
Author Response

(The authors gave the same response as above.)

Round 3
Reviewer 3 Report
Dear authors, your work has been improved significantly. However, there needs to be a clear boundary between Methodology and Results. In the Methodology, you should explain how you carried out the research, while in the Results, you should mention what you obtained once the Methodology was applied.
Put both parts in separate sections, i.e., 4. Methodology, 5. Results, and 6. Conclusions
Once you do this, your work will have a clearer structure.
Best wishes
Author Response
Please refer the attachment. Thank you very much for your kind suggestions and helpful comments.
